# Apocynin-Tandospirone Derivatives Suppress Methamphetamine-Induced Hyperlocomotion in Rats with Neonatal Exposure to Dizocilpine

**DOI:** 10.3390/jpm12030366

**Published:** 2022-02-27

**Authors:** Takashi Uehara, Masayoshi Kurachi, Takashi Kondo, Hitoshi Abe, Hiroko Itoh, Tomiki Sumiyoshi, Michio Suzuki

**Affiliations:** 1Department of Neuropsychiatry, Kanazawa Medical University, Uchinada 920-0293, Japan; 2Department of Neuropsychiatry, Graduate School of Medicine and Pharmaceutical Sciences, University of Toyama, Toyama 930-0194, Japan; uehara_takasayu@yahoo.co.jp (M.K.); mugen@med.u-toyama.ac.jp (H.I.); suzukim@med.u-toyama.ac.jp (M.S.); 3Center for Low-Temperature Plasma Sciences, Nagoya University, Nagoya 464-8603, Japan; tasakondo@khc.biglobe.ne.jp; 4Department of Applied Chemistry, Faculty of Engineering, University of Toyama, Toyama 930-8555, Japan; abeh@eng.u-toyama.ac.jp; 5Department of Preventive Intervention for Psychiatric Disorders, National Institute of Mental Health, National Center of Neurology and Psychiatry, Tokyo 187-8551, Japan; sumiyot@ncnp.go.jp

**Keywords:** apocynin, tandospirone, antioxidant, MK-801, locomotion, animal model, schizophrenia

## Abstract

Accumulating evidence implicates oxidative stress as a potential pathophysiological mechanism of schizophrenia. Accordingly, we synthesized new chemicals using apocynin and tandospirone as lead compounds (**A-2**, **A-3** and **A-4**). These novel compounds decreased reactive oxygen species (ROS) concentrations in vitro and reversed decreases in glutathione levels in the medial prefrontal cortex of rats transiently exposed to MK-801, an *N*-methyl-d-aspartate receptor antagonist, in the neonatal period. To determine whether **A-2**, **A-3** and **A-4** show behavioral effects associated with antipsychotic properties, the effects of these compounds on methamphetamine (MAP)-induced locomotor and vertical activity were examined in the model rats. **A-2** and **A-3**, administered for 14 days around the puberty period, ameliorated MAP-induced hyperlocomotion in MK-801-treated rats in the post-puberty period, while **A-4** suppressed MAP-induced vertical activity. These findings indicate that apocynin-tandospirone derivatives present anti-dopaminergic effects and may alleviate psychotic symptoms of schizophrenia.

## 1. Introduction

Redox dysregulation, *N*-methyl-d-aspartate (NMDA) receptor hypofunction and neuroinflammation may represent an aspect of the pathophysiology of schizophrenia [1]. For example, accumulating evidence has indicated increased oxidative stress in patients with schizophrenia and its prodromal phase [2,3]. Animal model studies have reported that NMDA receptor hypofunction in early life stages induces oxidative stress that causes reduction in the number of parvalbumin (PV)-positive γ-aminobutyric acid (GABA) interneurons in the prefrontal cortex (PFC) [4].

Decreased levels of redox substrates, especially glutathione (GSH), have been observed in cerebrospinal fluid (CSF), brain tissues, and whole blood of patients with schizophrenia [5]. Specifically, GSH levels are lower in the PFC of these patients, as measured by in vitro assay of postmortem brain tissues or proton magnetic resonance spectroscopy-based imaging [6,7]. GSH levels are also decreased in erythrocytes and plasma of patients with first-episode psychosis [8,9]. Moreover, erythrocyte GSH levels in individuals at ultra-high risk state have been reported to predict transition to psychosis [10]. These findings indicate that impairments in the GSH system may occur early in the illness trajectory and continue into the chronic phase.

A randomized controlled trial [11] found that N-acetylcysteine (NAC), an antioxidant and precursor of GSH, alleviated positive symptoms in patients with early phase schizophrenia, which was accompanied by an increase in GSH levels in the medial PFC. On the other hand, treatment with NAC prevented amphetamine (AMPH)-induced hyperlocomotion in a mice model of schizophrenia [12,13] that were exposed to social isolation rearing [14]. These findings suggest that antioxidants may ameliorate and/or prevent the hypersensitivity, reflecting increased dopamine (DA) transmissions in the limbic areas, through correcting GSH deficiency.

We synthesized new chemicals using apocynin and tandospirone as lead compounds [15], and have reported the ability of these chemicals to decrease reactive oxygen species (ROS) induced by X-irradiation in human lymphoma U937 cells [15]. Moreover, treatment with these compounds for 14 days around puberty ameliorated the decrease in GSH (mainly reduced GSH) levels and the reduction in PV-positive neurons in the medial prefrontal cortex (mPFC) of rats transiently exposed to MK-801, a highly potent and selective noncompetitive NMDA receptor antagonist, in the neonatal period [15]. These findings indicate antioxidant properties of the apocynin-tandospirone derivatives that normalize the GSH system.

The aim of this study was to determine whether apocynin-tandospirone derivatives elicit antipsychotic properties using an animal model of schizophrenia. For this purpose, we administered these compounds repeatedly to rats transiently exposed to MK-801 in the neonatal period, and examined methamphetamine (MAP)-induced locomotor and vertical activity. As reference compounds, clozapine (CLZ) and olanzapine (OLA) were selected, because these antipsychotic drugs are most commonly used to treat positive and negative symptoms of schizophrenia [16], and have been shown to protect against oxidative stress [17,18,19].

## 2. Materials and Methods

### 2.1. Animals and Neonatal MK-801 Treatment

Animal models were prepared based on previous reports [20,21,22,23]. Female Wistar rats obtained at 14 days of pregnancy (Japan SLC, Shizuoka, Japan) were individually housed at 24 ± 2 °C under a 12 h light–dark cycle (7:00–19:00) with free access to food and water. On PD 7, male pups (7–15 g) were randomly divided into two groups; they received MK-801 (dizocilpine, 0.20 mg/kg, s.c.; Sigma–Aldrich, St. Louis, MO, USA; MK-801 neonatal treatment group) or an equal volume of saline (control; saline group) once daily for 4 days. Pups in each group came from at least 2 independent litters and received injections between 8:00 and 10:00. After weaning at PD 21, pups were group-housed by treatment (4–6 animals per cage with free access to food and water). The procedures complied with the National Institutes of Health guide for the care and use of Laboratory animals. All experiments were reviewed and approved by the Committee of Animal Research, University of Toyama.

### 2.2. Drug Administration

CLZ and OLA were purchased from Wako Pure Chemical Industries Co. (Osaka, Japan). npApocynin-tandospirone derivatives (**A-2**, **A-3**, **A-4**; Figure 1) were synthesized in our laboratory [15]. They were dissolved in saline and were administered s.c. at 2.5 mg/kg. CLZ or OLA were administered at 5.0 mg/kg (s.c.) and 0.2 mg/kg (s.c.), respectively. An equal volume of saline was used as a vehicle. Animals were assigned to one of the following 12 groups: saline-saline group (*n* = 12), MK801-saline group (*n* = 12), saline-A2 group (*n* = 12), MK801-A2 group (*n* = 13), saline-A3 group (*n* = 12), MK801-A3 group (*n* = 12), saline-A4 group (*n* = 13), MK801-A4 group (*n* = 13), saline-CLZ group (*n* = 12), MK801-CLZ group (*n* = 12), saline-OLA group (*n* = 12), and MK801-OLA group (*n* = 12). All drugs or saline were administered once daily (8:00–10:00) for 14 days (PD 43–56).

### 2.3. Locomotor Activity Testing

Locomotor activity was tested on PD 57 (24 h after the last drug administration). Locomotion was measured in an ambulation observation chamber (blackened vinyl chloride cages, 40 cm × 40 cm × 40 cm; AMB-3001, Ohara & Co., Ltd., Tokyo, Japan) equipped with 6 × 6 photoelectric light sources spaced at 7 cm intervals and 2.5 cm (for horizontal locomotion) above the floor (AMB-2020, Ohara & Co., Ltd.) [20,22,24]. Vertical activity was measured using photoelectric light sources spaced 19 cm above the floor [22]. Rats were brought to the testing room in their home cages and were immediately placed in the test chamber. Spontaneous activity was measured for 15 min. Then, each rat was administered (s.c.) 1.0 mg/kg MAP (3.0 mg/mL; Dainippon Sumitomo Pharmaceuticals, Tokyo, Japan) 30 min after placed in the test chamber, and MAP-induced activity was recorded for 90 min. Interruptions of light beams were registered as activity counts and were summarized every 5 min by the Logger Interface control system (IF-10-LOG, Ohara & Co., Ltd.) (Figure 2).

### 2.4. Presentation of the Results and Statistics

Data were analyzed by SPSS software (version 19.0 J for Mac, SPSS Japan Inc., Tokyo, Japan). For comparison of spontaneous locomotor activity and vertical activity, two-way ANOVA was performed with neonatal treatment status (treatment: MK-801 and saline) and drug administration (drug, **A-2**, **A-3**, **A-4**, CLZ, OLA and saline). For comparison of MAP-induced locomotor activity and vertical activity, activity counts were obtained for every 15 min. Repeated measures ANOVA with treatment status (treatment: MK-801 and saline) and drug administration (drug: **A-2**, **A-3**, **A-4**, CLZ, OLA and saline) as between-subject factor, and time as within-subject factor was treated as repeated-measures variable. When appropriate, repeated-measures ANOVA were performed for each drug administration (**A-2**, **A-3**, **A-4**, CLZ, OLA vs. saline) separately, because our a priori hypothesis predicted the effect of apocynin-tandospirone derivatives on hyperlocomotion induced by neonatal MK-801 treatment. Moreover, this was followed by one-way ANOVA and post hoc Bonferroni tests were used to evaluate between-group differences at each time. A probability (p) of less than 0.05 was considered to be significant.

## 3. Results

### 3.1. Spontaneous Locomotion

There was no significant treatment × drug interaction or main effect of neonatal treatment status or drug administration (Figure 3).

### 3.2. MAP-Induced Locomotion

There were drug × time interactions (F(30,810) = 2.26, *p* < 0.001) and main effects of drug (F(5135) = 3.03, *p* = 0.013). These results demonstrated that there were different effects on MAP-induced hyperlocomotion between drug administration. Next, repeated- measures ANOVA were performed for each drug administration (**A-2**, **A-3**, **A-4**, CLZ, OLA vs. saline) separately. Administration of **A-2**, **A-3** or OLA ameliorated increased MAP-induced locomotor activity in MK-801 treated animals (Figure 4A,B,E). There were significant treatment × drug × time interactions (F(6270) = 2.68, *p* = 0.015) and treatment × drug interactions (F(1,45) = 6.64, *p* = 0.013) with **A-2**. Significant treatment × drug interaction with **A-3** and with OLA (**A-3**; F(1,44) = 4.64, *p* = 0.037, OLA; F(1,44) = 5.54, *p* = 0.023, respectively).

Next, we sought to determine if the effect of **A-2**, **A-3** and OLA on MAP-induced hyperlocomotion in MK-801 treated rats would be affected by time. The augmentation of an increase in MAP-induced locomotor activity in MK-801 treated animals was significantly suppressed from 16 to 45 min with **A-2** and 1 to 45 min with **A-3** after MAP injection (Ps < 0.05, one-way ANOVA followed by Bonferroni test), whereas OLA did not decrease MAP-induced hyperlocomotion at each time.

**A-4** had no effect on MAP-induced locomotor activity in the rats treated with neonatal MK-801 (Figure 4C). On the other hand, there were significant drug × time interaction (F(6264) = 2.17, *p* = 0.047) and significant main effects of treatment (F(1,44) = 5.65, *p* = 0.022) and drug administration (F(1,47) = 6.17, *p* = 0.017) with CLZ (Figure 4D). This result indicated that CLZ increased locomotor activity by itself.

### 3.3. MAP-Induced Vertical Activity

Repeated measures ANOVA demonstrated a significant drug × time interaction (F(30,810) = 1.75, *p* = 0.008). This result showed that there were different effects on MAP-induced vertical activity between drug administration. Then, repeated-measures ANOVA was conducted to examine drug administration effects of each drug (**A-2**, **A-3**, **A-4**, CLZ, OLA vs. saline) on MAP-induced vertical activity separately. There was treatment × drug interaction (F(1,46) = 5.04, *p* = 0.03) with **A-4** (Figure 5C). Administration of **A-2**, **A-3** or OLA did not affect MAP-induced vertical activity (Figure 5A,B,E). CLZ intensified MAP-induced vertical activity (drug administration × time interaction, F(6264) = 2.58, *p* = 0.019) (Figure 5D).

## 4. Discussion

This study demonstrated that **A-2**, **A-3** and OLA, but not **A-4**, suppressed MAP-induced hyperlocomotion in rats exposed to neonatal MK-801. Inhibition of hyperlocomotion by **A-2** and **A-3** occurred in the early phase (within 45 min) after MAP administration. On the other hand, only **A-4** ameliorated increased MAP-induced vertical activity. By contrast, spontaneous locomotor and vertical activities were not affected by any of the test drugs, suggesting that **A-2** and **A-3** show antidopaminergic properties in the model rats (Table 1).

### 4.1. Effects of New Chemicals and Atypical Antipsychotics on MAP-Induced Hyperactivity

The results of this study are consistent with the observations that NMDA antagonists, such as phencyclidine (PCP) and MK-801, during the early developmental stage causes long-term alterations in behavioral activity in rodents [20,22,25,26]. Since MAP-induced hyperlocomotion is closely related to enhanced dopamine release in the nucleus accumbens (NAC) [27], blockade of NMDA receptors in the neonatal stage may lead to exaggerated dopamine transmissions in this brain region.

**A-2** and **A-3** significantly inhibited MAP-induced hyperlocomotion in MK-801-treated rats, especially immediately after MAP administration, indicating that repeated treatment with **A-2** or **A-3** attenuates MAP-induced hyperdopaminergic states. These findings are in line with the observations which indicates repeated treatment with apocynin, a lead compound of **A-2** and **A-3**, attenuates the hyperlocomotion induced by MAP [28]. Moreover, similar behavioral changes were also observed at post-puberty, but not pre-puberty [20,22], indicating blockade of NMDA receptors between PD 7 and 10 causes delayed emergence of hyperdopaminergic states. In fact, this is reminiscent of the manifestation of positive symptoms of schizophrenia at the time of sexual maturation [29]. The ability of **A-2** or **A-3** to attenuate the behavioral abnormalities suggests that these compounds would potentially prevent the development of psychosis in subjects with vulnerable traits. On the other hand, **A-4** did not suppress MAP-induced hyperlocomotion in model rats, while it attenuated MAP-induced vertical activity. As this type of exploratory behaviors (e.g., rearing) are governed by mesolimbic and mesocortical dopaminergic transmissions [30,31,32], **A-4** may also elicit anti-dopaminergic effects in these dopaminergic pathways.

In this study, OLA suppressed MAP-induced hyperlocomotion in MK-801 treated rats, whereas CLZ intensified MAP-induced locomotor and vertical activities. These observations are in line with the ability of chronic treatment with OLA to alleviate AMPH-induced hyperlocomotion in sensitized mice [33]. By contrast, repeated administration of CLZ does not attenuate MAP-induced hyperlocomotion [34,35] or reverse AMPH-induced hyperlocomotion [36]. 

The ability of acute CLZ or OLA administration to inhibit AMPH-induced hyperlocomotion is considered to be mediated by the antagonistic action on dopamine D2 receptors [34]. However, the inhibitory effect of CLZ, but not OLA is gradually weakened by repeated administration [34]. Acute administration of AMPH increases free radical (H_2_O_2_) formation and decreases GSH levels in the striatum of rats [37]. Five-day treatment with AMPH induces extensive oxidative stress and long-lasting glial reactivity in the PFC [38]. Moreover, long-term use of AMPH or MAP injures both dopaminergic and serotonergic neurons, which is mediated, in part, by oxidative stress [39]. Therefore, reversal of MAP- or AMPH- induced hyperlocomotion in sensitized animals by chronic OLA treatment may be associated with protection against oxidative stress. On the other hand, chronic treatment with CLZ have been reported to reduce GSH levels in the PFC of naïve- [40] or MK-801-treated rats [15]. Thus, in the model rat studied here, CLZ augmented dopamine transmissions, possibly by attenuating activities of the GSH antioxidative system.

It is noteworthy that the behavioral profile of **A-4** is OLZ-like, and different from those of **A-2** or **A-3**, although three compounds were synthesized from common leading compounds. **A-4**, among the three compounds, has been found to most effectively scavenge intracellular ROS formation induced by X-irradiation in human lymphoma U937 cells [15]. In addition, all compounds (**A-2**, **A-3** and **A-4**) reversed the decrease in GSH concentrations in the mPFC of neonatal MK-801 treated rats, whereas OLA did not affect them [15]. These findings suggest that the effect of three compounds on MAP-induced hyperactivity in model animals were caused by not only anti-oxidative property, but also antagonistic effects on DA and/or serotonin (5-HT) receptors. Further studies are warranted to confirm the pharmacological profile of apocynin-tandospirone derivatives, especially affinities for DA-D_2_ and 5-HT_2A_ receptors.

### 4.2. Mechanisms Underlying the Antidopaminergic Effects of Apocynin-Tandospirone Derivatives

Antidopaminergic effects of apocynin-tandospirone derivatives may be related to phamacologic properties similar to those of the mother compounds. Thus, repeated treatment with apocynin for 7 days dose-dependently decreased MAP-induced locomotor activity and dopamine release in the dorsal striata of rats [28], whereas such effects were absent with acute administration [28]. Therefore, the ability of apocynin to suppress MAP-induced dopamine release and hyperactivity may not totally be due to DA-D_2_ receptor blockade, but partly to suppression of oxidative stress as a NOX inhibitor [28]. Support for this speculation comes from observations that perinatal transient blockade of NMDA receptors accelerates apoptotic cell death in the hippocampus [41]. Additionally, animals with neonatal excitotoxic lesions of the hippocampus have been shown to elicit increased MAP-induced locomotion after puberty [42,43]. Accordingly, we previously observed that **A-2**, **A-3** and **A-4** decrease ROS activities in vitro, and that 14-day treatment with these agents ameliorated disturbances of the GSH system in the animal model used here [15]. 

Acute treatment with 5-HT_1A_ agonists, e.g., tandospirone and buspirone, has been shown to increase DA release in the mPFC and NAC of naïve rats [44,45,46]. This effect is blocked by systemic administration or local application of WAY 100635, a selective 5-HT_1A_ receptor antagonist [45]. Although tandospirone has a weak antagonistic effect on DA-D_2_ receptors (Ki: 1.7 µM) [47], these findings suggested that tandospirone-induced increase in DA levels in the mPFC is due to stimulation of 5-HT_1A_ receptors [45]. On the other hand, (±)-8-hydroxy-2-(di-n-propylamino)-tetralin hydrobromine (8-OH-DPAT), a selective 5-HT_1A_ agonist, inhibited AMPH-induced increases in extracellular DA levels in the mPFC and NAC [48,49]. The inhibitory effect of 8-OH-DPAT was completely blocked by WAY 100635. Moreover, 8-OH-DPAT diminished the magnitude of footshock-induced increase in DA utilization in the mPFC [50]. Therefore, 5-HT_1A_ agonism may have differential effects on tonic and phasic DA transmissions in the limbic areas. It is worth noting that SEP-363856, a novel psychotropic agent that lacks D_2_ receptor affinity, alleviates psychotic symptoms of schizophrenia in a placebo-controlled randomized clinical trial [51]. Importantly, SEP-363856 dose-dependently inhibits PCP-induced hyperactivity responses in C57BL/6J mice, which was partially attenuated by pretreatment with WAY-100636 [52]. As PCP increases DA release in the PFC and NAC [53], it is possible that apocynin-tandospirone derivatives suppress increased MAP-induced locomotion and vertical activity through 5-HT_1A_ receptor stimulation.

Because this study showed the effect of chronic treatment with the novel compounds on MAP-induced hyperactivity in the animal model of schizophrenia, the current data may not directly explain the mechanisms underlining the antidopaminergic effects of apocynin-tandospirone derivatives. Further studies are warranted to clarify the acute effects of the new drugs on MAP-induced hyperactivity and to confirm the affinity for various receptors of these compounds, especially DA-D2, 5-HT_1A_ and 5-HT_2A_ receptors.

## 5. Conclusions

Results of the current study demonstrate that treatment with apocynin-tandospirone derivatives **A-2** and **A-3** around puberty mitigates increased MAP-induced hyperlocomotion in rats transiently exposed to NMDA blockers in neonatal periods. These findings suggest some of the apocynin-tandospirone derivatives, through antioxidant effects [15], may provide a novel strategy for early intervention and prevention of schizophrenia and related conditions. Specifically, agents to modify redox dysregulation may ameliorate psychotic symptoms and prevent the development of psychosis in vulnerable subjects [3].

## Figures and Tables

**Figure 1 jpm-12-00366-f001:**
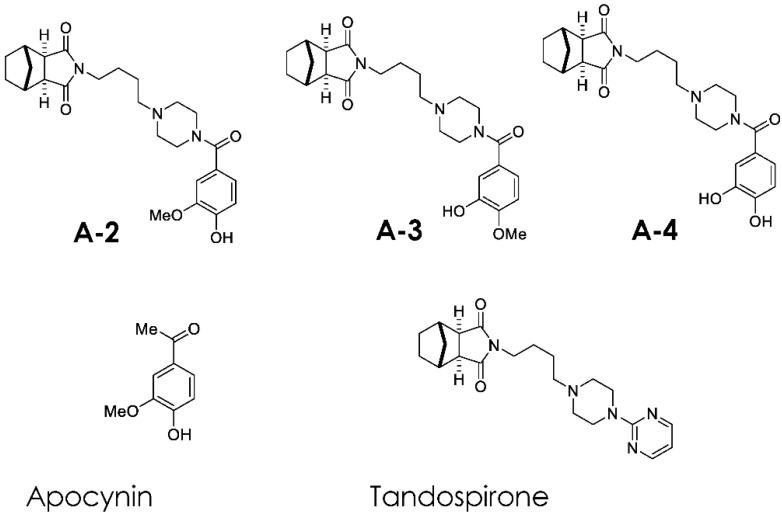
Chemical structures of three apocynin-tandospirone derivatives (**A-2**, **A-3**, and **A-4**) and their lead chemicals (apocynin and tandospirone).

**Figure 2 jpm-12-00366-f002:**
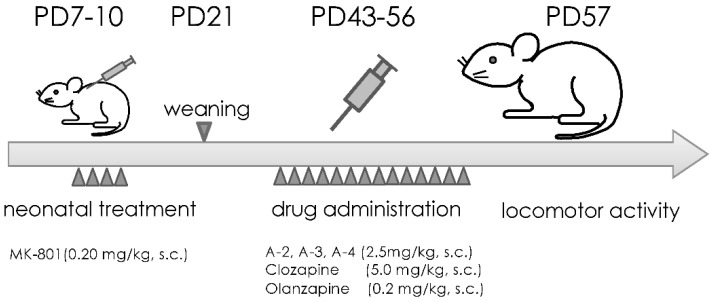
Schematic representation of experiments for neonatal treatment (MK-801), drug administration (**A-2**, **A-3**, **A-4**, clozapine and olanzapine) and locomotor activity testing. PD, postnatal days.

**Figure 3 jpm-12-00366-f003:**
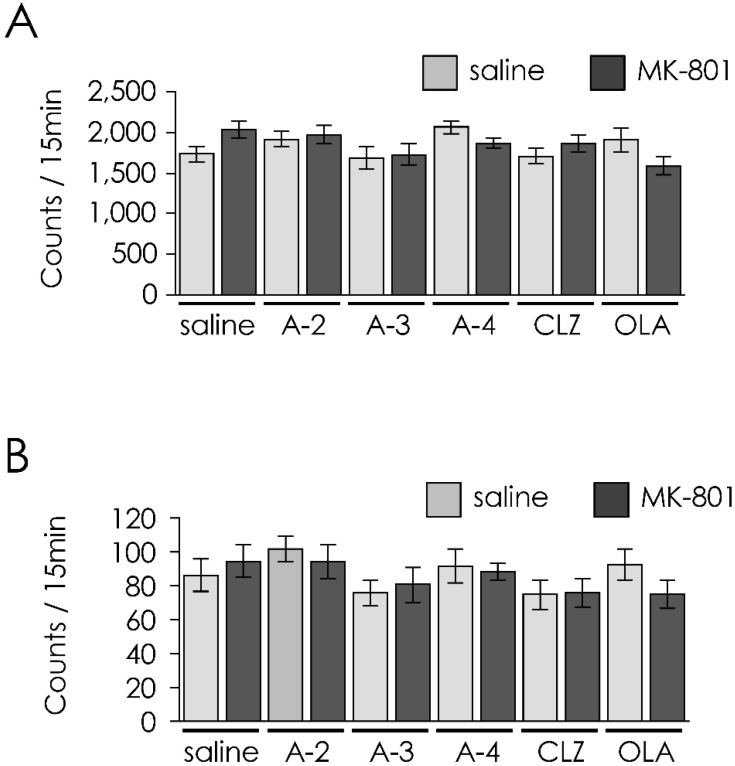
Effect of 14-day (PD 43–56) treatment with apocynin-tandospirone derivatives and antipsychotics on spontaneous (**A**) locomotor activity and (**B**) vertical activity (for 15 min after placed in the test chamber) on PD 57. Saline-saline group (*n* = 12), MK801-saline group (*n* = 12), saline-A2 group (*n* = 12), MK801-A2 group (*n* = 13), saline-A3 group (*n* = 12), MK801-A3 group (*n* = 12), saline-A4 group (*n* = 13), MK801-A4 group (*n* = 13), saline-CLZ group (*n* = 12), MK801-CLZ group (*n* = 12), saline-OLA group (*n* = 12), and MK801-OLA group (*n* = 12). Values are expressed as means ± SEM.

**Figure 4 jpm-12-00366-f004:**
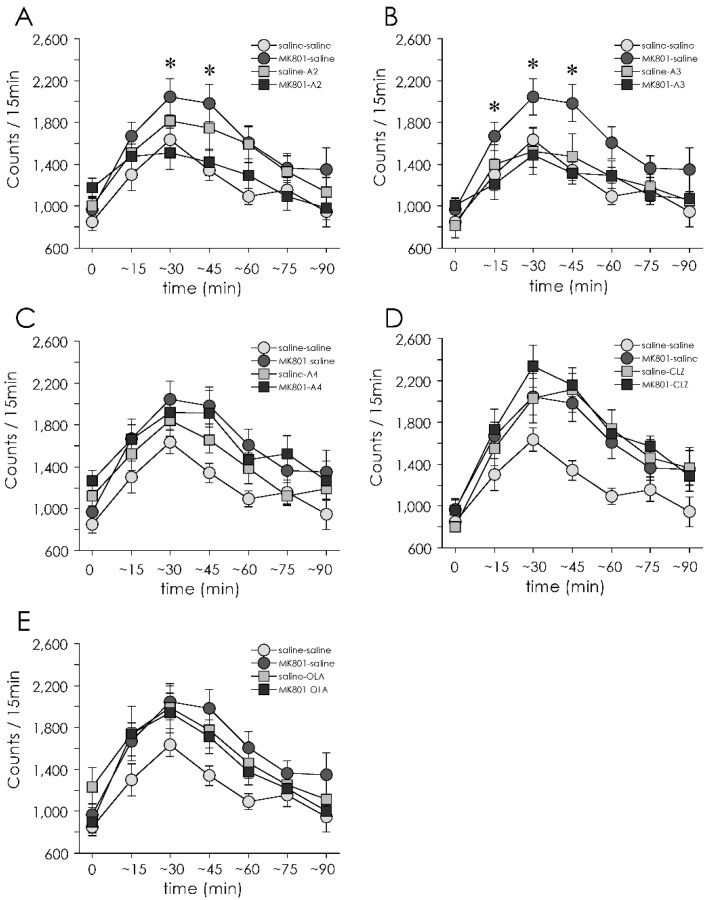
Effect of 14-day (PD 43–56) treatment with apocynin-tandospirone derivatives and antipsychotics on MAP-induced locomotor activity (for 90 min) on PD 57. Data with **A-2** (**A**), **A-3** (**B**), **A-4** (**C**), CLZ (**D**) and OLA (**E**) are shown. Saline-saline group (*n* = 12), MK801-saline group (*n* = 12), saline-A2 group (*n* = 12), MK801-A2 group (*n* = 13), saline-A3 group (*n* = 12), MK801-A3 group (*n* = 12), saline-A4 group (*n* = 13), MK801-A4 group (*n* = 13), saline-CLZ group (*n* = 12), MK801-CLZ group (*n* = 12), saline-OLA group (*n* = 12), and MK801-OLA group (*n* = 12). Values are expressed as means ± SEM. * *p* < 0.05 compared MK-801-**A-2** or **A-3** with MK-801-saline group.

**Figure 5 jpm-12-00366-f005:**
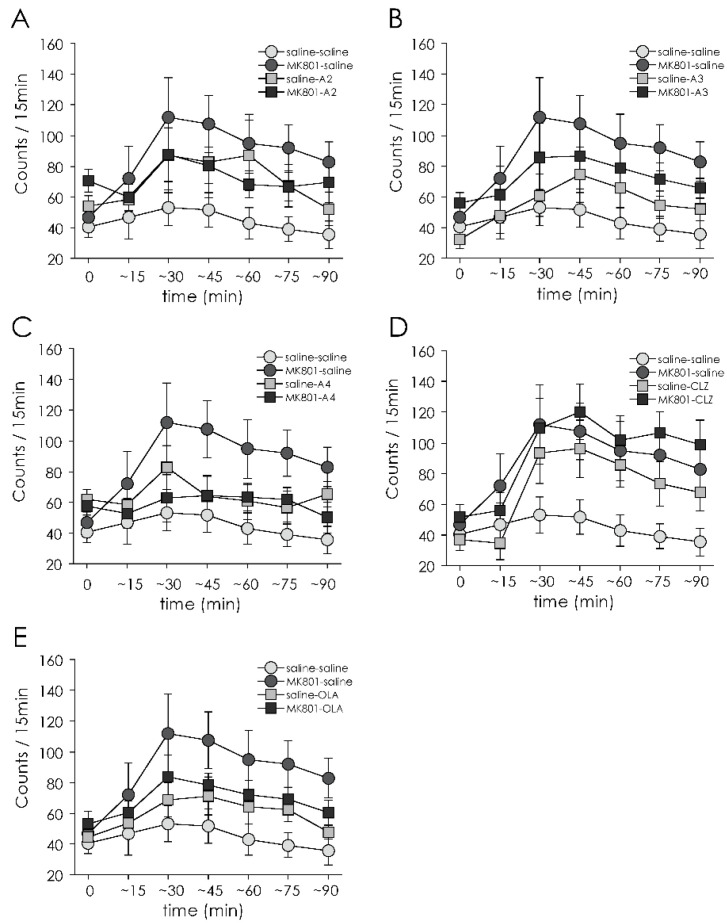
Effect of 14-day (PD 43–56) treatment with apocynin-tandospirone derivatives and antipsychotics on MAP-induced vertical activity (for 90 min) on PD 57. Data with **A-2** (**A**), **A-3** (**B**), **A-4** (**C**), CLZ (**D**) and OLA (**E**) are shown. Saline-saline group (*n* = 12), MK801-saline group (*n* = 12), saline-A2 group (*n* = 12), MK801-A2 group (*n* = 13), saline-A3 group (*n* = 12), MK801-A3 group (*n* = 12), saline-A4 group (*n* = 13), MK801-A4 group (*n* = 13), saline-CLZ group (*n* = 12), MK801-CLZ group (*n* = 12), saline-OLA group (*n* = 12), and MK801-OLA group (*n* = 12). Values are expressed as means ± SEM.

**Table 1 jpm-12-00366-t001:** Effects of apocynin-tandospirone derivatives and antipsychotic drugs on locomotor and vertical activity.

	Spontaneous	MAP-Induced
	Locomotor Activity	Vertical Activity	Locomotor Activity	Vertical Activity
**A-2**	−	−	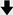	−
**A-3**	−	−	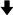	−
**A-4**	−	−	−	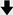
CLZ	−	−	−	−
OLA	−	−	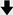	−

Arrow demonstrates suppressed effect of drug on MAP-induced hyperactivity. Minus sign “−” shows no change compared with spontaneous or methamphetamine-induced activity. MAP, methamphetamine; CLZ, clozapine; OLA, olanzapine.

## Data Availability

The data that support the findings of this study are available from the corresponding author upon reasonable request.

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
