# Peer review of "Apocynin-Tandospirone Derivatives Suppress Methamphetamine-Induced Hyperlocomotion in Rats with Neonatal Exposure to Dizocilpine"

_jpm, 2022, doi:10.3390/jpm12030366_

Round 1

Reviewer 1 Report

 Takashi Uehara et. al in manuscript titled “Apocynin-tandospirone derivatives suppress methamphetamine-induced hyperlocomotion in rats with neonatal exposure to dizocilpine” is significant manuscript and have found promising novel compounds which mitigated MAP-induced hyperlocomotion and suggest that these compounds may mitigate symptoms of schizophrenia. The manuscript is promising, and few suggestions might improve manuscript.

Major points:

  1. I suggest adding significant * or p-values in graph to see the significance. In addition, to that adding “n” to figure legends would be helpful (I see them in methods).
  2. Discussion can be more elaborate, and authors need to address the limitation of the study. What could be possible side effect of the drug which could be or could not be nulled with these novel drugs?

Author Response

reviewer 1

Takashi Uehara et. al in manuscript titled “Apocynin-tandospirone derivatives suppress methamphetamine-induced hyperlocomotion in rats with neonatal exposure to dizocilpine” is significant manuscript and have found promising novel compounds which mitigated MAP-induced hyperlocomotion and suggest that these compounds may mitigate symptoms of schizophrenia. The manuscript is promising, and few suggestions might improve manuscript.

  1. I suggest adding significant * or p-values in graph to see the significance. In addition, to that adding “n” to figure legends would be helpful (I see them in methods).

answer: According to reviewer’s suggestion, we have added the information of number of each group in figure legends (Fig.3, 4, 5)

  1. Discussion can be more elaborate, and authors need to address the limitation of the study. What could be possible side effect of the drug which could be or could not be nulled with these novel drugs?

answer: As suggested by reviewer, we have added the limitation of the study in the last in “Mechanisms underlying the antidopaminergic effects of apocynin-tandospirone derivatives” section, as follows (p.11, line 27-32). We agree that side effect of the drugs is important, further studies are warranted to examine the toxicity of these drugs.

Because this study showed the effect of chronic treatment with the novel compounds on MAP-induced hyperactivity in the animal model of schizophrenia, the data from the current study may not directly explain the mechanisms underlining the antidopaminergic effects of apocynin-tandospirone derivatives. Further studies are warranted to clarify the acute effects of the new drugs on MAP-induced hyperactivity and to confirm the affinity for various receptors of these compounds, especially DA-D2, 5-HT1A and 5-HT2A receptors.

Reviewer 2 Report

 The manuscript  is organised , clear, relevant for the field .

The hypothesis is well defined and the methods, the design configuration conducts to the clear results. The results are presented in tables and figures appropriates, easy to follow.

The discussions shows the compare data with the literature, clear.

The conclusions are sustained by the results. 

The references are mostly recent. 

The authors have the etics statements.

Author Response

reviewer 2

The manuscript is organized, clear, relevant for the field.

  1. The hypothesis is well defined and the methods, the design configuration conducts to the clear results. The results are presented in tables and figures appropriates, easy to follow.

answer: According to the reviewer’s suggestion, we have added the table which showed summary of results in this study (p.9, line 7).

  1. The discussions show the compare data with the literature, clear.

answer: According to reviewer’ suggestion, we have added sentence comparing the data with the literature, as follows (p.9, line 18-20).

These findings are in line with the observations which indicates repeated treatment with apocynin, a lead compound of A-2 and A-3, attenuates the hyperlocomotion induced by MAP [28].

  1. The conclusions are sustained by the results.

answer: According to reviewer’s suggestion, we have added the conclusion sustained by the results in the head in “Conclusion’ section, as follows (p.12, line 2-4).

Results of the current study demonstrate that treatment with apocynin-tandospirone derivatives A-2 and A-3 around puberty mitigate increased MAP-induced hyperlocomotion in rats transiently exposed to NMDA blockers in neonatal periods.

  1. The references are mostly recent.

answer: According to the reviewer’s suggestion, we have updated the references which we could do it (p.9, line 28).

  1. The authors have the ethics statements.

answer: According to the reviewer’s suggestion, we have added the ethics statements in “2.1. Animals and Neonatal MK-801 Treatment” section, as follows (p.5, line 10-12).

The procedures complied with the National Institutes of Health guide for the care and use of Laboratory animals. All experiments were reviewed and approved by the Committee of Animal Research, University of Toyama.
